# Pre-Service Science Teachers' Understanding of Socio-Scientific Issues Instruction through a Co-Design and Co-Teaching Approach Amidst the COVID-19 Pandemic

**Mingchun Huang [1],\* and Peng He [2],\*** 

1    College of Teacher Education, Capital Normal University, Beijing 100037, China
2    College of Education, Michigan State University, East Lansing, MI 48824, USA
\*    Correspondence: hmc91810@126.com (M.H.); hepeng1@msu.edu (P.H.)

**Abstract:** This qualitative case study explores the development of pre-service science teachers' (PSTs) understanding of Socio-Scientific Issues (SSI) instruction. The study utilized a conceptual framework of SSI-based instruction and a three-phase approach involving co-designing solutions, co-designing curriculum materials, and co-teaching classes. Primary data sources included PSTs' interviews and reflective journals, while artifacts, field notes, and curriculum materials served as secondary data sources. Thematic analysis was used to analyze the data of six PSTs in a teacher preparation program. We found that the PSTs' understanding of SSI instruction was enhanced in 12 features of three core aspects: design elements, learner experiences, and teacher attributes. "Engaging in higher-order practices" was the most prominent feature, observed across all three phases. The three-phase approach played a crucial role in promoting PSTs' understanding of SSI instruction, with each phase guiding their understanding in distinct ways. Particularly, the co-designing solutions phase facilitated the development of PSTs' "awareness of the social considerations associated with the issue". While the co-designing curriculum materials phase was effective in promoting their understanding of "scaffolding for practice: providing scaffolding for higher-order practices. "The co-teaching classes phase played a crucial role in facilitating their understanding of "willingness to position oneself as a knowledge contributor rather than the sole authority".

**Keywords:** pre-service science teachers; SSI instruction; co-design and co-teaching

## 1. Introduction

In recent years, United Nations documents have emphasized the value of education for sustainable development by highlighting its role in cultivating environmentally responsible citizenship and social sustainability [1–4]. Education for Sustainable Development (ESD) is becoming an educational approach that fosters critical thinking, transformative learning, and societal transformation for a sustainable, inclusive, and responsible world [4]. Teachers, as influential change agents, are vital for achieving Sustainable Development Goals through their knowledge and competencies in transforming educational processes towards sustainability. Teacher education must shift towards ESD by adopting proven and innovative methods, establishing training programs, and furthering efforts to help future teachers understand the integration of environmental protection and personal development within educational processes [3]. Incorporating socio-scientific issues (SSI) into educational curricula and emphasizing education's role in addressing sustainable development goals are crucial for preparing future teachers and citizens [5–7]. By integrating socio-scientific issues (SSI) into the learning experiences of students and pre-service science teachers (PSTs), we can nurture the critical thinking and problem-solving abilities required to create sustainable and resilient societies. Global challenges, such as the COVID-19 pandemic, have underscored the complexities and urgencies of incorporating SSI into science education and science teacher education [8–10]. This crisis has not only amplified the difficulties

in integrating SSI into pedagogical practices but also emphasized the significance of SSI instruction in fostering PSTs' learning and responsible citizenship among students. In light of the pandemic, it is imperative for educators and scholars to develop innovative approaches to address complex real-world dilemmas within this rapidly evolving context, ultimately enhancing the relevance and efficacy of SSI integration.

SSI-based instruction is a pedagogical approach that aims to engage students in science learning by involving them in investigations of SSIs that relate to their lives and communities [6,11]. This approach focuses on the application of scientific knowledge to solve real-world problems, with the goal of developing students' abilities to think critically, make informed decisions, and communicate their ideas effectively. SSI-based instruction also emphasizes the importance of interdisciplinary learning, collaboration, and active engagement with scientific knowledge and practices [12]. Overall, SSI-based instruction aims to prepare students for active participation in democratic decision-making and to foster a scientifically literate citizenry [6,13]. The field of science teacher education encounters two major challenges for PSTs' professional learning as follows: (a) the procedural nature of pedagogical knowledge and the apprentice-style approach to teacher education, and (b) limited time for learning a large amount of content [14]. The first challenge results in PSTs having a good understanding of educational theory but struggling to apply it in practice. The second challenge refers to the difficulty that PSTs face in mastering innovative pedagogy (e.g., SSI instruction) within a short timeframe. These challenges have informed the need for designing and implementing science teacher education programs to support their professional learning.

There is an increasing body of research on SSI instruction and PSTs' professional learning. Much of the literature focuses on investigating the impact of a specific SSI instruction program [15–17] or intervention on the development of PSTs to help them incorporate SSI into their future classrooms [18–20]. The findings in previous research suggest that PSTs face significant barriers to understanding and implementing SSI instruction. They tend to be apprehensive about SSI, preferring the familiarity and controllability of teaching specific subject matter, and are inclined to avoid the complexity and uncertainty of SSI teaching that emphasizes contextualization, socio-cultural and interdisciplinary connections, and activities such as deliberation, argumentation, and inquiry [15]. PSTs lack confidence in both the content and pedagogy of SSI instruction [7,21], which makes it difficult for them to understand SSI and provide effective teaching feedback when designing and implementing SSI instruction. Besides, the existing research indicates that participating in the design and implementation process of SSI curriculum units helps transform PSTs' understanding of SSI and engage them in critical practices [22]. In particular, participating in a community of practice during SSI implementation also promotes teacher professional development [23]. Some research advocates that teacher education programs should provide more opportunities for PSTs to experience SSI instruction [20].

Overall, SSI instruction remains an auxiliary component of pre-service teacher education, appearing in the form of specific programs or projects. There is a lack of discussion about professional learning models to support PSTs' SSI instruction. Therefore, it is necessary to explore how to design PST professional learning models that promote understanding of SSI instruction. This study aims to examine PSTs' understanding of SSI-based instruction through three phases, including co-design solutions, co-design of curriculum materials, and co-teaching in SSI-based classrooms. Our research question: What is the PSTs' understanding of SSI instruction when they experienced the co-design solutions and curriculum materials, and co-teaching phases?

## 2. Conceptual Framework

### 2.1. Co-Design and Co-Teaching: Collaborative Approaches of Professional Learning for Preservice Teachers

Collaborations among teacher education researchers and in-service and pre-service teachers have positive impacts on PSTs' professional development in their understand-

ing of teaching and confidence [24]. Research indicates that establishing a collaborative relationship between teachers and researchers, especially in more complex forms of teaching, such as SSI instruction, helps teachers develop their professional knowledge of these instructional approaches [25]. Collaborative efforts among PSTs, in-service teachers, and researchers are more effective in promoting the PSTs' development than one-to-one mentoring relationships between PSTs and their mentors [26,27].

The co-design of curriculum materials has been regarded as an effective way for teachers' professional learning [28–30]. The co-design approach also can enhance teachers' in-depth understanding of teaching and scientific content [30,31]. In this study, we regard this approach as a professional learning approach that researchers and teachers collaborate as a community to develop and refine curriculum materials. However, simultaneous curriculum design and teacher professional learning, even for in-service teachers, is a demanding task [30]. Teachers face pressure and need to strike a balance between the learning of both content and pedagogical knowledge. Therefore, when facilitating PSTs through their participation in co-design curriculum, it is necessary to consider their characteristics, such as the weakness of subject content knowledge and insufficient pedagogical knowledge.

Co-design solutions refer to a process of working collaboratively to develop and refine potential solutions in solving real-world problems. Design solutions were emphasized as one of eight scientific and engineering practices in the Framework of K-12 Science Education [32]. Designing solutions is essential for scientists and engineers when they meet a specific problem or challenge. In the context of preservice science teacher education, "co-design solution" provides opportunities for PSTs to work collaboratively with mentors, experienced teachers, and peers to design solutions. The co-design solution is therefore positioned as an independent phase that PSTs must experience, especially nowadays to address problem solving in real-world situations. PSTs should experience such a design solution process by collaborating with others so that they obtain such problem-solving knowledge and skills for designing SSI-based curriculum and instruction for classroom teaching. However, previous studies suggest that PSTs often lack content knowledge and confidence in teaching SSI, which results in negative attitudes toward these issues [33]. By engaging PSTs in co-design solutions, they can deepen their understanding of science content and develop a more nuanced understanding of the challenges and complexities of teaching SSI [22]. This, in turn, helps increase their confidence and capabilities to teach SSIs in classrooms. Such an approach could support PSTs' content knowledge and skills, as well as their capabilities to collaborate and communicate with others. Therefore, this study includes the co-design solution phase for our PSTs before they start designing the SSI curriculum.

Collaborative curriculum design provides opportunities for teachers to think about teaching in new ways, yet substantial changes in thinking and practice can only occur if teachers utilize the materials they develop and critically reflect on their current instructional practices [34]. Co-teaching is a relational practice in which two or more teachers plan, teach, and assess students together, offering a valuable tool for enhancing teachers' professional skills [35]. Co-teaching, particularly between cooperating or mentoring teachers and PSTs, provides vital support to help PSTs develop their professional competencies [36,37]. Researchers have investigated the potential benefits of co-teaching for PSTs and found that it provides a rich and shared experience that fosters professional conversations among peers, cooperating teachers, and university researchers [38]. For example, Gallo-Fox and Scantlebury Ref. [39] found that co-teaching was an effective method for developing teaching skills and knowledge among PSTs in a secondary science teacher education program. Guise, M.; Habib, M.; Thiessen, K.; etc. Ref. [27] further suggested that co-teaching offers valuable opportunities for collaboration, reflection, and professional development for PSTs. Furthermore, PSTs' understanding of SSI instruction is deeply influenced by their field experiences, and co-teaching with cooperating teachers in secondary schools, which is beneficial for both PSTs and in-service teachers. Therefore, co-teaching represents an effective approach to supporting PSTs' professional development.

### 2.2. SSI Instruction: A Framework for Socio-Scientific Issues-Based Education

This study employs a framework for guiding PSTs' understanding of SSI instruction, consisting of three key aspects: design elements, learner experiences, and teacher attributes [13]. Design elements are the essential aspects of an instructional design that influence how SSIs are presented to students. Learner experiences are the activities and interactions that students engage in as they learn about SSIs. Teacher attributes are the characteristics and practices of effective teachers who are skilled in SSI instruction. In particular, the aspect of design elements includes four features originally: (a) building instruction around a compelling issue, (b) presenting the issue first, (c) providing scaffolding for higher-order practices, and (d) providing a culminating experience. In this study, we combined the first two elements into a single element, "Issue First: Building instruction around an authentic issue and presenting it first". In addition, we added a new design element, "SSI Teaching and Learning Sequence", which is based on the SSI Teaching and Learning Framework [40]. All other features in the other two aspects (learner experiences and teacher attributes) remain unchanged in the framework of SSI instruction [13]. Thus, each of the three aspects contains four essential features (see Table 1).

**Table 1.** Essential features in the three components of the SSI instruction framework [13,40].

| Design Elements | Learner Experiences | Teacher Attributes |
| --- | --- | --- |
| 1. Issue First: Building instruction around an authentic issue, presenting it first<br>2. T&L Sequence: SSI teaching and learning sequence<br>3. Scaffolding: Scaffolding for practice: providing scaffolding for higher-order practices<br>4. Solution: Providing a culminating experience (develop a position or solution) | 1. HO Practice: Engaging in higher-order practices (problem-solving)<br>2. SI&T: Confronting scientific ideas and theories related to the issue<br>3. Data Collection: Collecting and/or analyzing scientific data related to the issue<br>4. Negotiate: Negotiating social (e.g., political and economic) dimensions | 1. Know Content: Knowledgeable about the science content related to the issue<br>2. Social Considerations: Aware of the social considerations associated with the issue<br>3. Position Self: Willing to position self as a knowledge contributor rather than sole authority<br>4. Uncertainties: Willingness to deal with uncertainties in the classroom |

Drawing on the existing research, along with the framework of SSI instruction, we articulate a three-phase approach that encompasses co-designing solutions, co-designing curriculum materials, and co-teaching classes, to support PSTs' understanding of SSI instruction (See Figure 1). By engaging in collaborative efforts in our professional learning program, we expect PSTs can achieve a profound understanding of the three aspects of SSI instruction through their experiences in the co-design solution (Phase 1), co-design of the curriculum materials (Phase 2), and co-teaching with the designed curriculum materials in class (Phase 3).

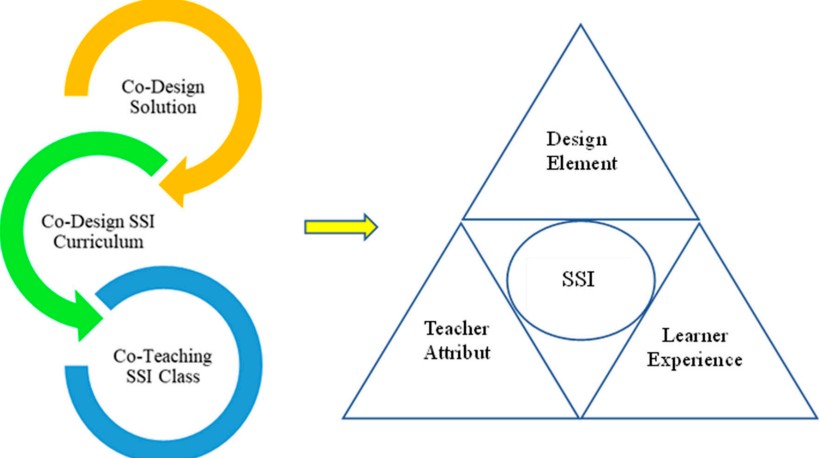

**Figure 1.** A three-phase approach to support PSTs' understanding of SSI instruction. (the right side [13] is modified from Presley et al., 2013).

## 3. Methods

### 3.1. Background

During the 2021 summer and fall semesters, the PST course in our university's School of Teacher Education faced a hybrid learning environment that shifted between online and in-person due to the COVID-19 pandemic. Both the practical internships and teaching method courses for PSTs were affected. The first author, serving as the PSTs' mentor, assisted her mentees in enhancing their teaching abilities in secondary schools and stimulating their interests in continuous learning of teacher preparation courses during the pandemic. Meanwhile, a local teacher requested to collaborate with the first author in conducting a teaching research project as she was planning to offer an elective course for her high school students in the upcoming fall semester. The in-service teacher expressed the collaborative efforts to develop new curriculum materials to enhance students' interest and proficiency in learning chemistry. Therefore, the first author intended to form a collaborative team, including teacher educators, in-service teachers, and pre-service teachers to develop a flexible and practical professional learning community for PSTs. Through co-designing SSI solutions, co-designing SSI curriculum materials, and co-teaching SSI classes, one purpose of this professional learning program aims to help PSTs adapt better to current and future educational contexts.

In science education, SSI education is regarded as having the ability to blend emotional power and scientific content, which in turn advances the development of students' scientific literacy [23,41,42]. The COVID-19 pandemic is a palpable social reality, and as science educators, we aspire to view this "obstacle" as an opportunity to apply our scientific knowledge to address societal problems. During that exceptional period, we also noted that COVID-19 had presented numerous SSIs that we had experienced firsthand [43,44]. For instance, in the midst of the COVID-19 outbreak, many individuals and families underwent stress and panic due to the scarcity and hoarding of disinfectants. As individuals with a background in science education, we resolved to concentrate on this SSI for inquiry.

PSTs as future teachers face the challenge of improving their understanding and ability to teach SSI. The pandemic presents an opportunity for students to engage in many SSIs to deepen problem-solving practice as well as gain insight into the value of science and technology. In China, during the pre-pandemic time, the shortage of disinfectants in the market affected many families. Thus, the COVID-19 pandemic serves as an uncertain environment in this study. The SSI in this study lies in: How to help people reduce the anxiety caused by the shortage and snapping up of disinfectants in the market during the pandemic? Thus, we had our PSTs engaged in the entire process of co-designing solutions, co-designing curriculum materials, and co-teaching toward the SSI. This study aims to explore PSTs' understanding of SSI instruction across the three phases.

### 3.2. Participants

The study focuses on six PSTs who participated in the entire research process with the teacher educators and in-service teachers. These PSTs were graduate students in chemistry education at a normal university (i.e., teacher education university) in Beijing, China. They did not have any prior formal teaching experience in secondary schools during this study. All of whom were in their first year of a two-year professional education master's program at the time of the study. The participants fulfilled all of the obligations of the PST professional development program during their non-formal university hours. Table 2 shows the information on six PSTs. All names of the six participants are pseudonyms.

**Table 2.** Information of six PSTs.

|  | Alice | Bob | Carol | Dan | Edith | Fred |
|---|---|---|---|---|---|---|
| Phase 1 | Syringe group | Spray Bottle group | Syringe group | Spray Bottle group | Cookie Box group | Spray Bottle group |
| Phase 2 | Lesson 3–4 designer |  | Lesson 1–2 designer |  | Lesson 5–6 designer |  |
| Phase 3 | Co-teaching lessons 1–6 |  |  |  |  |  |

### 3.3. Procedure

The research design from "define and design" to "prepare, collect data" for this study can be found in Figure 2. See "Data analyze procedure " in Figure 3.

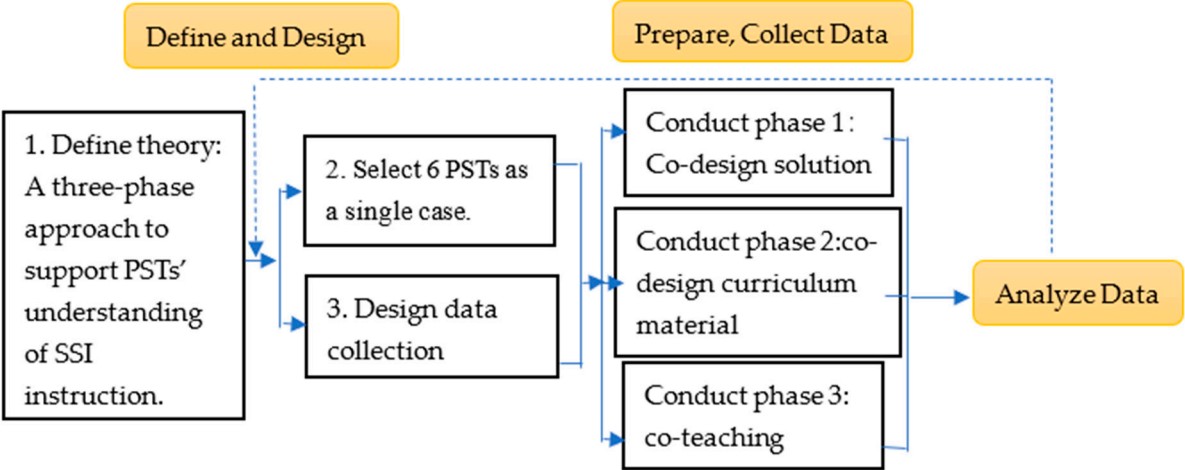

**Figure 2.** The research design for this study.

#### 3.3.1. Phase 1: PSTs as Solution Co-Designers to Address Socio-Scientific Issues

This phase aims to understand how PSTs would approach an SSI problem and engage in a problem-solving process. The process involved a collective discussion among the research team to identify a specific SSI related to the pandemic so that the team could address it with their current abilities and desired approach. Following this discussion, the research team focused on a real-world SSI topic: How can individuals alleviate stress and panic caused by the scarcity and hoarding of disinfectants during the pandemic? The team unanimously decided to address this SSI by "designing a simple disinfectant device for emergency use". On this basis, two background studies were conducted by the PSTs: (a) a thorough review of literature for exploring knowledge related to disinfectants and identifying relevant scientific content based on secondary school science standards, and (b) an investigation of disinfectant-related products through online shopping platforms, followed by purchases, disassembly, and experimentation. Next, the six PSTs were divided into three small groups to search for and utilize available materials in their surroundings. Each pair worked together on finding solutions and designing artifacts, including selecting raw materials, assembling devices, testing and implementing artifact functions, and closely collaborating with their partners. The teacher educators and in-service teachers provided online and offline consultation and advice. Finally, the three groups presented their products within the team, respectively. The research team collected data, including artifacts from the three groups, reflective journals from each PST, and field notes with photos and videos during the product creation process.

#### 3.3.2. Phase 2: PSTs as Co-Designers of Curricula for Teaching Socio-Scientific Issues

The research team guided the PSTs to enter the second phase of co-designing the SSI curriculum. They collaboratively developed a curriculum unit to solve the SSI problem of the scarcity of disinfectants on the market through the design and production of a disinfectant device for emergency use. This unit consists of six lessons (as shown in Table 3). The lessons were completed over three weeks in an elective course in high school, with two lessons taught each week and each session lasting 50 min. The strategy was to first allow the six PSTs to discuss with the teacher educators and in-service teachers, as well as among themselves, to jointly build the SSI unit sequence. The six lessons were respectively focused on orientation and planning (Lessons 1–2, design and improvement (Lessons 3–4), and presentation and feedback (Lessons 5–6). Next, the six PSTs paired with each other

to form three groups based on their interests and familiarity with the content. The first group was responsible for Lessons 3–4, which is about the design and production class. The second group was responsible for Lessons 1–2, aiming at the guiding planning class. The third group was responsible for Lessons 5–6, focusing on the reflective presentation class. After the first version of the unit was completed, the entire research team collaboratively discussed it, with the teacher educators giving suggestions for further revisions. Each group then made modifications to their first version of materials accordingly. Finally, the in-service teacher provided suggestions on their lessons from the perspective of actual implementation. After that, each group made their final revisions. Overall, the PSTs co-operated with the teacher educators and in-service teachers in designing and writing the SSI unit. During this phase, we collected the data, including six lessons for the first round and modified lessons. The research team also collected PSTs' reflective journals during this phase.

**Table 3.** Lesson sequence and activities in the SSI-based unit.

| Week | Lesson | Learning Activities |
| --- | --- | --- |
| 1. Orientation and planning | 1 | a. Scientific reading: Determining the Specification Demand of product design |
| | 2 | b. Experimental Demonstration: Experiencing a disinfector and Observing Design Prototypes |
| | | c. Group Collaboration: Planning of Projects and Designs solutions |
| 2. Design and improvement | 3 | a. Observational Analysis: Exploring the Scientific Principles and Technique in Use of Electrolysis |
| | | b. Product Design and Fabrication: Drawing Schematics and Initial Production |
| | 4 | c. Experimental investigation: iterative improvement of the disinfector |
| 3. Presentation and feedback | 5 | a. Group Presentation and Display of Designed Products |
| | 6 | b. Students' feedback on learning |
| | | c. Teacher summarizes the whole learning process |

3.3.3. Phase 3: PSTs as Co-Teachers in Socio-Scientific Issue (SSI) Classrooms

In the third phase of co-teaching, the three groups of PSTs provided an explanation and demonstration of the unit to the cooperating teacher prior to classroom teaching. The cooperating teacher was also informed of the time and location of lessons and the relevant student information. The elective course was implemented on Friday afternoons for three weeks during the fall semester of 2021 in an affiliated high school in Beijing. Thirty high school students were divided into five groups for cooperative learning. We then discussed the appropriate way for PSTs' co-teaching with the cooperating teacher as they did not have teaching experience and the students were high achievers at the school. We decided that the six PSTs serve as teaching assistants, independently choosing to join one of the student groups, guiding student learning activities, and serving as classroom observers to record student learning. During each lesson, they used their cell phones to record key student activities and prompted students to think deeply and provided immediate feedback for them to improve their work. After each lesson, they reflected on the lesson with the cooperating teacher and the teacher education researcher. After the final lesson, the first author conducted an overall interview with the six PSTs regarding their participation in the three-phase SSI-related process. The data collected in this phase includes all six PSTs' field notes based on classroom observations, reflective journals, and transcriptions of the final comprehensive interviews.

**4. Data Analysis**

The study is a case study [45] that explores the learning experiences of six PSTs about their understanding of SSI teaching in the three phases of our professional development program. This study employs two designed instruments as our primary data sources: first, a three-phase reflective journal that encompasses three questions addressing the gains, shortcomings, and expectations for assistance experienced by the pre-service teachers (PSTs)

in each phase. The second instrument is a final interview outline. The semi-structured interview was conducted at the end of Phase 3, with each PST interview lasting approximately 30–45 min. Questions within the latter instrument are formulated in accordance with the Socio-Scientific Issues (SSI) framework [13]. While the interviews allowed the PSTs to describe their participation in the three phases, their learning experiences, and their ideas about enacting each phase independently. One example question is: "What are your experiences in addressing SSI problems during the process of co-designing solutions?" This question is derived from the 'Design Elements' feature in the SSI framework, specifically from "4. Solution: Providing a culminating experience (develop a position or solution)" (see Table 1). In addition to these primary data sources, we used various secondary data sources, including PSTs' artifacts and field notes about the process, curriculum materials with two versions, classroom observation notes, and lesson videos. We employed the thematic analysis approach to examine the data, analyzing interviews (n = 6), their reflective journals on their learning experiences (n = 18) related to SSI instruction development in high school classes, as well as curriculum materials (n = 6), and field notes created by the six PSTs [46].

Our data analysis process consisted of two manuals coding rounds (see Figure 3). In the first round, two authors used deductive codes based on "A Framework for Socio-scientific Issues Based Education" [13] to code the multiple data sources for each of the 12 features of SSI instruction (see Table 1). In the second round, the data were coded into three components. To ensure the validity of the themes, we conducted the triangulation of data sources [47], which involved PSTs observing 18 classroom lessons and creating three modeling artifacts. To ensure the internal validity of our data analysis, the two coding processes were conducted independently by two researchers. The results obtained by each of them were compared, and any discrepancies were discussed until an agreement was reached. Ultimately, the researchers compiled the frequencies of each element obtained from the statistical encoding after two rounds of coding to examine whether the PSTs' understanding of SSI instruction had developed after experiencing all three phases. Additionally, they sought to identify the most prominent developmental points in the PSTs' understanding of SSI instruction within each phase, in order to ascertain the value of each phase in promoting their understanding of SSI instruction.

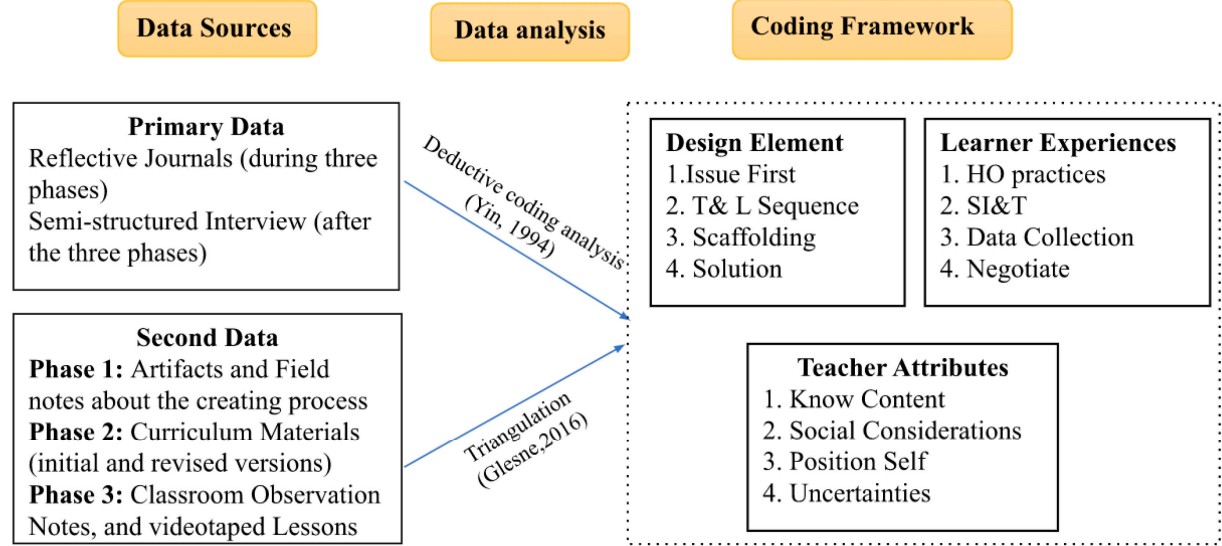

**Figure 3.** Data analysis procedure [45,47].

## 5. Findings

*5.1. PSTs Exhibit a Comprehensive Understanding of SSI Instructions through the Three-Phases Approach*

Based on the data analysis, we found that the PSTs' understanding of SSI instruction was enhanced in twelve elements of three core aspects: design elements, learner experiences, and teacher attributes.

PSTs' understanding of "learner experiences" is demonstrated in several ways based on their interview and reflective journals. Firstly, they engage in higher-order practices such as using a variable power supply to adjust the voltage by observing the rate of bubble production on the electrodes to produce sodium hypochlorite. As one of PSTs, Edith mentioned that the variable power supply can also investigate the effect of voltage on the amount of sodium hypochlorite produced. Secondly, they confronted scientific ideas and theories related to the issue of disinfectants, such as understanding the preparation and testing principles of disinfectants and the working principle of disinfectors, as Bob mentioned. Thirdly, they collected and analyzed scientific data related to the issue, such as searching and organizing literature on disinfectants and determining commonly used disinfectants for emphasis. Finally, they negotiated social dimensions, such as using chemical knowledge to analyze and distinguish rumors and make rational decisions. For example, Bob mentioned that those with chemical knowledge need to determine which disinfectant is better or if they are equivalent by considering their properties and disinfection principles when encountering rumors about disinfectants, such as chlorine dioxide.

The PSTs' understanding of "Design elements" is centered around "Building instruction around an authentic issue, presenting it first", which involves extracting the driving force behind a project from social issues and selecting preparation principles according to practical needs as Edith learned. Additionally, the design of the device should be based on existing products, and the product should be prepared based on actual usage. The "SSI Teaching and Learning sequence" emphasizes using the SSI Teaching and Learning sequence when writing curriculum materials, starting from a broad perspective and gradually narrowing down to the preparation of specific disinfectants and the production of simple disinfectors, which two PSTs (Edith and Carol) have also mentioned. The element of "Scaffolding for practice: Providing scaffolding for higher-order practices" will be further elaborated on later. "Providing a culminating experience" involved investigating the increased demand for products, such as masks and disinfectants, during the COVID-19 pandemic, reviewing the literature on disinfection and protection principles, and creating simple and effective products through disassembling actual products and using materials with similar functions found in daily life to replace product components.

PSTs demonstrated their understanding of "teacher attributes" in several ways. They were knowledgeable about science content related to the issue and emphasized interdisciplinary content in designing and creating SSI learning activities. They promoted hands-on work and considered multiple factors, strengthening connections between various disciplines, and enabling students to understand material laws and changes from multiple perspectives. For instance, Fred and Bob chose disinfection principles based on the need for effective prevention of COVID-19 in households. They understood the social considerations associated with the issue and are willing to position themselves as knowledge contributors rather than sole authorities as previously stated. Additionally, PSTs are willing to deal with uncertainties in the classroom, which will be discussed further in the following section.

*5.2. PSTs Demonstrated Significant Gains in "Engaging in Higher-Order Practices" across the Three Phases*

The PSTs' understanding of the "Engaging in higher-order practices (problem-solving)" aspect was notable and consistent across all three phases.

### 5.2.1. Co-Design Solution: PSTs' Realization of the Importance of High-Order Practices in Addressing SSI

The PSTs realized the importance of high-order practices, specifically problem-solving, in addressing SSI. They gained valuable experience in the design process of making disinfectors. They encountered problems, experienced failures, and continually improved their process of making disinfectors. Through this process, PSTs understood the complexity of problem-solving and the importance of problem-solving competencies in addressing SSI situations. PSTs also realized that SSI requires engagement in high-order scientific practices, which involves not only solving scientific problems but also dealing with problems in society.

On one hand, they constantly considered the use of scientific concepts and technical devices to acquire design solutions. For instance, Dan reported that "*the resulting amount of sodium hypochlorite was too small to be measured with starch potassium iodide test paper*". After discussing with team members, they concluded that "*the battery power provided was insufficient to prepare the disinfectant quickly, which ultimately led to the unsuccessful result*". Meanwhile, Fred stated that "*various issues were identified, such as the lack of product due to poor battery electrode contact and high electrical energy loss caused by the choice of pencil core for the electrode*", but these problems were eventually solved through "*incremental improvements, resulting in the successful production of sodium hypochlorite*". These experiences demonstrated the PSTs' abilities to troubleshoot and iteratively improve their solutions, which is essential in addressing complex societal issues.

On the other hand, the PSTs had an in-depth understanding of the complexity of real problem-solving and the basic principles and ideas for solving problems. For instance, Fred stated that "*in real-life situations, problems are often complex and involve multiple factors and a broad range of knowledge, so the first step is to simplify*". When dealing with COVID-19, various aspects, such as the virus structure, mask protective effects, and disinfectants, can effectively kill the virus, which should be considered. Fred also emphasized that "*focusing on a specific aspect within the larger context allows us to search for information more effectively*". Another PST, Carol reported that when faced with social problems, "*it is important to first clarify the problems to be solved and the desired results, and then prioritize these problems*". She added that "*one should consider the possible solutions, weigh the pros and cons, and seek help and cooperation from others. During the problem-solving process, one should pay attention to any changes and adjust if necessary*". Afterward, reflecting on the experience and shortcomings provides a better response to similar problems in the future.

### 5.2.2. Co-Design Curriculum: PSTs Considered "Engaging in Higher-Order Practices" as Key Learning Activities for Design in Curriculum Materials

In the second phase, PSTs considered which scientific practices should be selected for student engagement, and they presented particular activities, ordered practice activities, and promoted students' development.

PSTs considered carefully which scientific practices to engage students in when designing curriculum materials for "design and create" courses. For instance, Edith summarized that the curriculum content for such courses should include three main parts. The first part involves obtaining the necessary materials by selecting appropriate ones, finding the raw materials for their preparation, and selecting the right preparation method based on how the substance is made in the laboratory. The second part revolves around understanding real products, including interpreting the product and planning a project. The third part, focused on creating and improving products, includes analyzing the results of others' projects, creating the product, releasing it, and reflecting on it. PSTs' consideration of these three parts of curriculum content ensures that students engage in the appropriate scientific practices for a well-rounded and effective educational experience.

When designing the learning activities, they considered the specific presentation of a particular practice activity. This is exemplified by the "Communication and Discussion" activity in the disinfectant section (see Figure 4). As described by Carol, real disinfectant

images, videos, and product information were extracted for this activity to provide students with a preliminary understanding of the working principle and structure of a simple disinfectant. By using this as a starting point for divergent thinking, students were able to write down their own ideas and thoughts on making a simple disinfectant. This approach highlights the importance of selecting appropriate resources and materials to engage students in a meaningful and effective way.

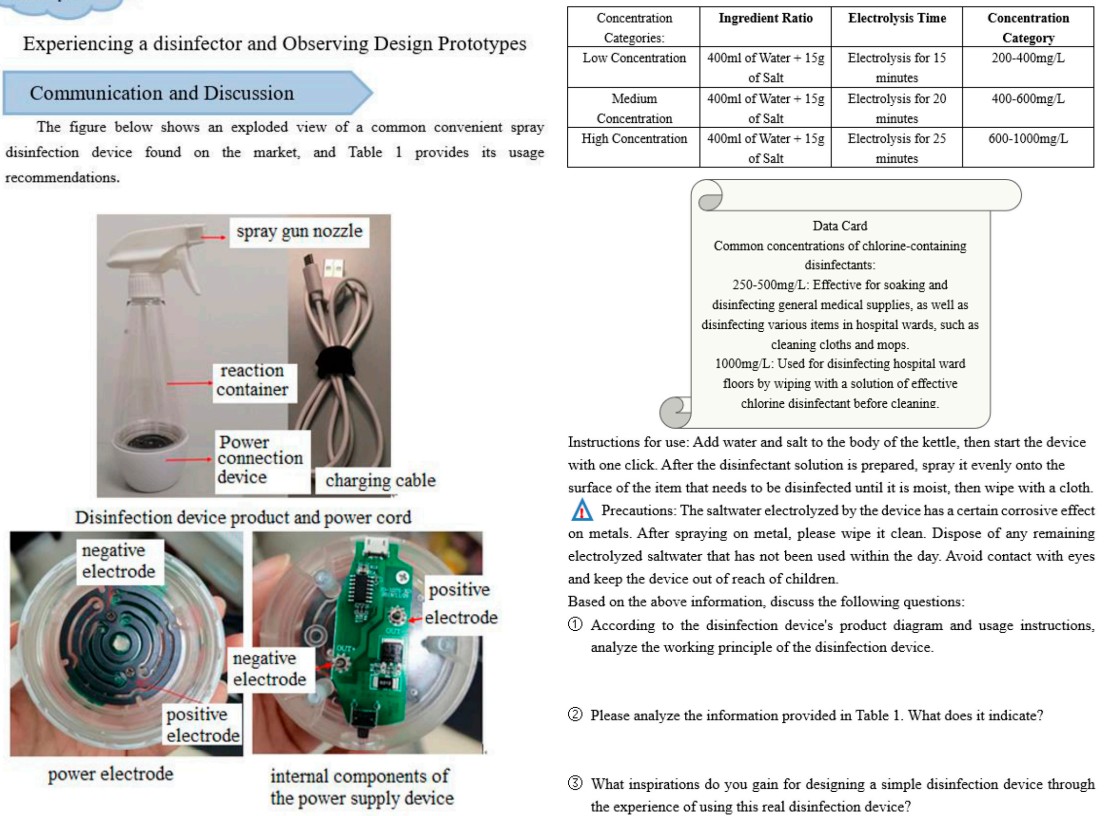

**Figure 4.** Carol's co-designing curriculum material with learning activity 2b "Communication and Discussion".

PSTs recognized the importance of arranging the order of practice activities and promoting students' development. As Edith stated, "*The learning activities should be designed in a progressive manner, and we should take into consideration students' knowledge of chemistry, physics, and other related disciplines. Starting from the desired outcome, the necessary conditions should be analyzed step by step*". The PSTs realized that the learning activities should progress in a logical order that leads to the final outcome. In the arrangement of textbook activities, it is important to start from the objectives of the production tasks and proceed step-by-step to analyze the necessary conditions. Firstly, appropriate disinfectants should be sought, followed by the selection of suitable devices and the exploration of appropriate conditions to ultimately produce the final product. It is also crucial to include a reflection and improvement stage, which encourages students to consider the strengths and weaknesses of their own products, explore and attempt to improve their homemade products, develop a general approach to problem-solving, and cultivate the habit of reflection and improvement.

### 5.2.3. Co-Teaching in SSI Class: PSTs Focus on Students' Process and Difficulties in Engaging in Higher-Order Practices, and Reflect on Feasible Teaching Strategies

When referring to "Engaging in higher-order practices", the PSTs had a greater number of observations on how students became involved in the problem-solving process and acquired more strategies for teaching problem-solving.

PST's observation highlights the problem-solving process that students engage in during SSI instruction. In exploring the problem of power supply, Edith observed the students' problem-solving process, which involved connecting the batteries in series to augment the voltage and successfully integrating the batteries with the reaction apparatus (see Figure 5).

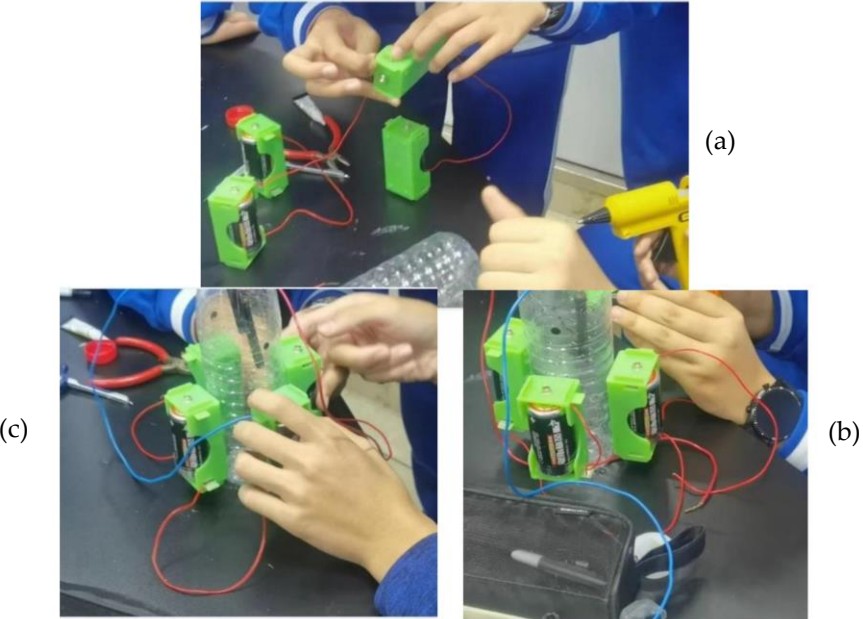

**Figure 5.** Edith's field note picture of high school students' problem-solving process. (**a**–**c**) indicate the order in which to view the images.

PST enhanced their understanding of students' problem-solving processes. In the classroom field note, Fred wrote that "While observing high school students' production process, my group initially placed the electrodes at the bottom of the bottle but experienced a water leakage. When I asked why the electrodes had to be placed at the lower part of the bottle, my classmates considered the issue of electrode contact area and believed that the greater the contact area between the electrode and the solution, the better the electrolysis effect. Later, they improved the plan by inserting the electrodes from the upper part of the bottle and passing the connecting wires through a hole in the upper part of the bottle to connect to an external power source. However, they also realized the importance of controlling the water level and leaving a certain volume of space when adding water, as hydrogen gas is produced during the electrolysis process, which could create excessive pressure and potentially rupture the bottle. Their approach was more comprehensive and thoughtful". [the classroom observation field note by Fred].

PSTs reflected on their curriculum material design and teaching strategies based on their observations of students' problem-solving abilities. "*When positioning the electrodes, the students in my group were also careful to avoid contact between the two electrodes, as this could result in a short circuit. It is evident that they considered several factors that could affect the electrolysis efficiency (concentration of sodium hypochlorite), such as electrode contact area, electrode spacing, and the corrosive nature of the solution after electrolysis. This was either due to their prior experience with the negative impact of these factors during product development or their extracurricular knowledge and understanding of these factors. This led me to consider whether the learning activities related to the "disinfectant device creation" could be enriched and deepened, such*

*as exploring the factors affecting the concentration of sodium hypochlorite (electrolysis efficiency)*".
[the classroom observation field note by Edith].

PSTs gained valuable insights into their strategies and approaches by observing and documenting the process of student problem-solving.

*5.3. PSTs Experienced Specific Growth in Their Understanding of SSI Instruction in Each of the Three Phases*

The three-phase approach played a crucial role in promoting PSTs' understanding of SSI instruction in distinct ways in each phase.

5.3.1. Phase 1-PSTs' Enhanced Awareness of Social Considerations Associated with SSI

In Phase 1, PSTs demonstrated the most significant improvement in their understanding of "Social considerations: Awareness of the social considerations associated with the issue". The study reveals that the co-design solution phase offers PSTs an opportunity to gain a deeper "awareness of the social considerations associated with addressing societal issues" such as the COVID-19 pandemic. To be specific, we found the following three aspects of their understanding in addressing such issues, including social policy awareness, market and resource accessibility, and science and social responsibility.

Firstly, PSTs learned about social policy awareness which emphasizes the significance of social policy knowledge in guiding actions taken to address societal issues, such as COVID-19 prevention. As such, relying solely on book knowledge is inadequate. A comprehensive understanding of the latest social policy developments is necessary to address societal issues effectively. For example, Alice stated, "*To effectively address various societal issues, including COVID-19 prevention through precise measures following Category B control implementation, it's crucial to also be aware of the latest societal developments and requirements for addressing such problems*". By gaining a deep understanding of social policy, PSTs became more responsible and aware of social considerations associated with various societal issues, such as the COVID-19 pandemic.

Secondly, PSTs understood the importance of market and resource accessibility in determining the accessibility and affordability of disinfectant products. "Market and resource accessibility" refers to the accessibility of disinfectants in the market, as well as the availability of resources for creating disinfectants using common household items in emergency situations. It further highlights the importance of investigating problems that are meaningful and engaging to students and connecting them with social development and productive life. Although there are various types of disinfectants available in the market, only a few can be prepared using common household items, which highlights the importance of having access to resources in emergency situations. Therefore, they utilized daily and readily available materials and the principle of electrolyzing saltwater in creating three distinct artifacts. Despite variations in design (see Figure 6), each artifact effectively demonstrated the PSTs' grasp and application of market and resource accessibility.

The *Spray Bottle* group     The *Cookie Box* group     The *Syringe* group

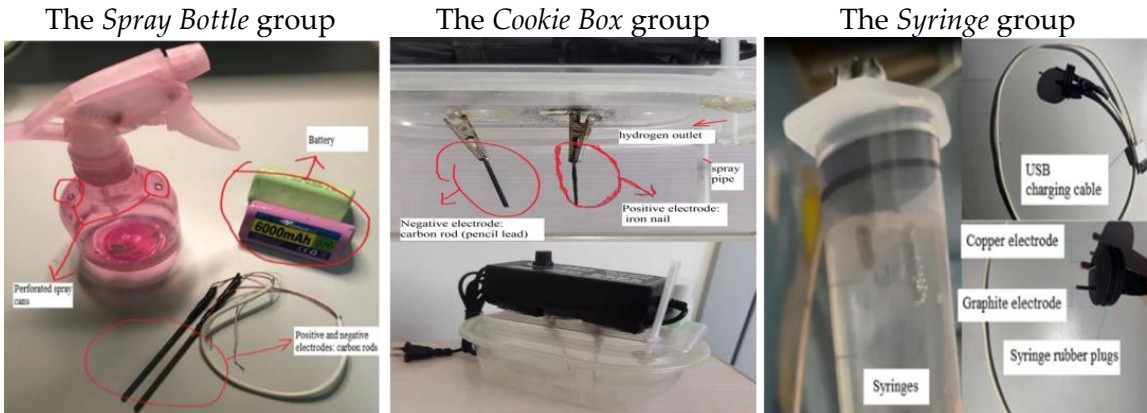

**Figure 6.** The artifacts designed by six PSTs.

Thirdly, PSTs developed their understanding of "science and social responsibility" by applying scientific knowledge and practices to develop solutions that meet social responsibilities. This term is significant in the context of using scientific knowledge and practices to meet social responsibilities during the COVID-19 pandemic. *For example, Bob* emphasized that "*the avoidance of immediate handling of the disinfectant in the bottle after preparation to prevent wastage, and to avoid pouring the disinfectant directly into the sewer to ensure the safety of the pipes*". PSTs realized that making homemade disinfectant can increase their knowledge, improve practical skills, and provide a sense of accomplishment, which helps ease the tension and anxiety caused by the pandemic.

5.3.2. Phase 2-PSTs' Improved Understanding of Scaffolding for Higher-Order Practices

In Phase 2, PSTs exhibited the most notable enhancement in their understanding of "Scaffolding for practice: Providing scaffolding for higher-order practices". The PSTs and research team worked together to construct a DST (Demand Specification, Scientific Principle, and Technique in Use, See Figure 7) thinking model to support students' thinking in the process of solving complex problems. They recognized the importance of continuously incorporating the DST model into the curriculum materials.

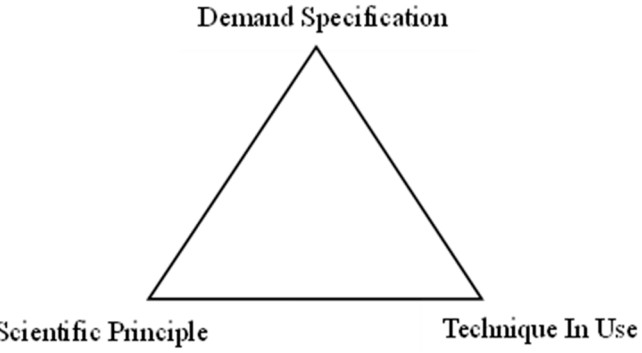

**Figure 7.** The DST thinking model.

Regarding the 1a learning activity (see Table 2) of the first lesson, which involved different types of disinfectants in scientific reading materials, both Edith and Bob noted the application of the thinking model. For instance, Bob indicated in the final version of the curriculum materials, "*based on the scientific reading section, we started with disinfectants and combined the scientific reading materials to ask students: 'What thoughts did you have about made a disinfectant after reading about disinfectants and their corresponding devices? 'The students' responses in the table included the demand specification, scientific principle, and technique in use as the three directions for thinking. This is not only the output of information from the reading materials but also implies the content that should be considered when preparing a disinfectant*".

Carol designed the 2b learning activity (see Table 2) of the first lesson to introduce students to disinfectant equipment by analyzing its operating principle through disassembly and reading instructions. This learning activity posed three questions, with the third question suggesting that teachers provide guidance from the perspectives of demand specification, scientific principle, and technique in use.

In addition, PSTs could think more critically and from multiple perspectives when it comes to developing and optimizing curriculum materials, thereby further promoting their understanding of scaffolding for higher-order practices. They recognized the importance of aligning materials with curriculum standards. For instance, Alice stated that "*when writing reading materials, I felt the need to explore how to use them in the classroom to achieve teaching goals. We should keep in mind the most critical elements such as textbooks and standards*". This indicates that PSTs have finally realized that standards serve as scaffolding for higher-order practices in the creation of curriculum materials and should be fully utilized.

PSTs learned to think about the guidance, accuracy, and progression of question design in the learning activities. For example, Bob stated that "*the materials should provide a clear*

*teaching direction, outline the required tasks, necessary materials, and questions that should be posed to students. The teacher's intended information to be obtained from the materials should be transformed into questions with a clear purpose and designed to guide both the students and the teacher"*. It was noted that question design should not be overly broad and should provide students with a clear direction for thinking. Another example, Dan noted that in curriculum materials, questions should possess a certain level of progression. For instance, a sense of progression is established by raising questions related to students' understanding and inspiration about disinfectants from the start. Additionally, he emphasized the importance of using appropriate wording when writing the curriculum materials, such as changing "*existing problems*" to "*discovered problems*" in the first lesson to better convey the material's meaning.

5.3.3. Phase 3-PSTs' Enhanced Understanding in Positioning Themselves as Knowledge Contributors Rather Than Sole Authorities

In Phase 3, PSTs displayed the most remarkable growth in their grasp of "Position self: Willingness to position oneself as a knowledge contributor rather than the sole authority". As they began to implement their own SSI curriculum in actual high school classrooms, they demonstrated a greater willingness to position themselves as knowledge contributors rather than sole authorities.

PSTs had a clearer sense of their multiple identities. In her reflection journal, Carol wrote that "PSTs, analyzed the performance of students and the teacher's teaching as a classroom observer". They then participated in the class as a teacher, where they found students' difficulties in understanding curriculum materials and how they solved problems, asking students "why they made such choices/designs/artifacts", particularly with regard to differences among group artifacts, and addressing questions that arise during student discussions. Additionally, they acted as a facilitator to hand out teaching aids and demonstrate products to support the cooperating teacher. Finally, PSTs acted to participate in the class as high school students, exploring what questions and ideas the students would have. Through these different roles, PSTs became more aware of their own multiple identities as observers, teachers, facilitators, and high school students, which enhances their abilities to understand and connect with students on different levels.

PSTs were willing to give more time to students, allowing them to present their work and learn from each other. According to Bob, during an "Artifacts presentation lesson", students should be given sufficient time to present their artifacts and evaluate each other. Students are also encouraged to make a presentation to explain their work and to introduce their artifact design ideas and advantages. The idea is to ensure that students have ample time to showcase their products and learn from each other's designs. This approach provides students with the opportunity to stand in front of the stage, recall and share their journey in the design and production process, and exchange and learn from each other. Bob believed that this method helps students gain a better understanding of the design and production process, fosters creativity and collaboration, and encourages them to take an active role in their learning.

After teaching in Phase 3, PSTs also reflected on their curriculum material design based on practical application. By considering the perspective of their students, PSTs could effectively develop the unit that facilitates the construction of scientific knowledge.

According to Fred, "In the first lesson, students' ideas on using disinfectants were scattered due to the fact that they had been exposed to too much extracurricular knowledge in a short period of time and had not paid close attention to their reading. PSTs noticed some students only focused on simple equations or principles they thought were easy to understand, but they were unable to apply them correctly. As a result, when asked why they chose a particular principle, most students were unable to provide a clear answer and were confused". PSTs began to consider their students' perspectives by matching the scientific reading materials with the content of the textbook and expanding on it. They also considered the current level of their students' knowledge and whether they could

understand the scientific content presented. The language and presentation of the materials were tailored to what the students comprehended. The materials were designed to progress from simple to complex, from familiar or unfamiliar substances, devices, and phenomena.

PSTs were aware of their own identity shift, as documented in Dan's reflective journal "Before starting the development task of the chlorine-containing disinfectant project, although I knew I should approach it from the perspective of a teacher and educator in project development and design, my subconscious was still stuck in the mindset of a student". This realization indicates a shift in the way PSTs perceive their role in the classroom, from a passive receiver of knowledge to an active contributor and facilitator of learning. "During the project development process, I unknowingly applied the same thinking and teaching techniques that were used by my high school teachers, and I even attempted to apply them in the project development". This awareness of their own learning experiences and how it influences their teaching practice can inform how PSTs approach curriculum design and pedagogy in the future.

Following their co-teaching of the SSI unit, PSTs shift towards a collaborative and contributive role in high school classrooms. PSTs exhibited a more refined and discerning understanding of their various professional identities and were more inclined to allocate a greater amount of class time for their students. PSTs were able to gain a deeper understanding of their roles as knowledge contributors rather than sole authorities. In this study, each of these three learning phases holds unique value in assisting PSTs in elevating their understanding of SSI instruction.

## 6. Discussion and Conclusions

This study employed a three-phase of co-design solutions, co-design curriculum materials, and co-teaching to develop PSTs' understanding of SSI instruction. We used the framework of SSI instruction [13,40] to analyze PSTs' multiple data and found that PSTs developed a deep understanding of SSI instruction through a three-phase approach.

The three-phase professional learning model provides an avenue for developing PSTs' understanding of SSI instruction. This study tackled the challenges described by existing literature [11] to develop PSTs' professional learning in teaching SSI. To address that, we expanded the models of co-design curriculum materials [34] and co-teaching [35] to a three-phase phase that includes a co-design solution phase for SSI problems [32] before the two phases. Moreover, our study also shows that the three phases are related to each other to support PSTs' understanding of SSI instruction. The empirical findings of this study support our argument that PSTs obtained the content and practices through their own problem-solving process, which could be also the foundations for their curriculum design and classroom teaching. PSTs' co-design curriculum experience could support them in better understanding SSI instruction before the co-teaching phase and deepen their reflection during the co-teaching phase. This study provides empirical evidence that PSTs could develop their understanding of SSI instruction by engaging in the three-phase professional learning process.

Each phase plays a critical role in promoting PSTs' understanding of SSI instruction. Addressing the deficiency of content knowledge and practices [15,23], we included the phase of co-designing solutions to support PSTs' understanding of SSI instruction. Our findings suggest that PSTs enriched their understanding of scientific knowledge and practices related to SSI problems-the shortage of disinfectants during the pandemic. Moreover, PSTs enhanced awareness of the social considerations associated with SSI in this phase. Existing research suggests that PSTs should engage in co-designing curriculum materials to enhance their in-depth understanding of teaching and scientific knowledge [29,30,48]. Consistent with those previous studies, we found that PSTs demonstrated a relatively good grasp of fundamental content knowledge, such as electrolysis and chlorine and its compounds. In addition, they also enriched their knowledge of chemical apparatus, general technical devices, and product manufacturing skills. Moreover, during the second phase, PSTs primarily developed various understandings related to instructional design. However, this process

mitigated the weaknesses of PSTs' subject content knowledge and insufficient pedagogical knowledge [14]. Regarding PSTs' development in the co-teaching phase, our findings are consistent with previous studies [35,36] that engaging in the co-teaching process helped PSTs gain teaching experience and transform their design into practice [34]. Moreover, we found that co-teaching could help PSTs understand their identities and become aware of transforming their multiple roles to better support student learning.

We conclude that this study presents a three-phase professional learning, consisting of co-design solutions, co-design curriculum materials, and co-teaching, to develop PSTs' understanding of SSI instruction. Each phase has a critical role in developing PSTs' understanding, especially in the aspects of design elements, learner experiences, and teacher attributes. The three-phase approach fully aligns with the key pedagogical approaches in ESD, which are "A learner-centered approach", "Action-oriented learning", and "Transformative learning" [4]. Based on research, we also realized that it is vital not only to include Sustainability-related content in the curricula but also to use action-oriented transformative pedagogy. The three-phase approach, proven to be valuable for PSTs' professional learning, offers insightful guidance for designing PSTs' teacher education activities. To master an instructional method, PSTs not only need to be taught relevant content but also need to engage in corresponding instructional practices, design appropriate curriculum materials, and immerse themselves in the method as if they were secondary school students experiencing this instructional method. However, some data in this study reveal that after the three-phase process, PSTs can "know" the uncertainty in SSI instruction, but concurrently, "the PSTs' tendency to control the class is a common observation". The potential inconsistency between "knowing" and "doing" among PSTs suggests that the challenges in pre-service teacher education are not solely rooted in content knowledge and pedagogical content deficiencies but also in the transformation of their learning approaches. The three-stage method in this study serves as an approach to promote PSTs' professional learning, but how they will act in future instruction still requires more refined support and further research evidence for full substantiation. Additionally, our findings cannot be generated in a large context or other disciplines due to the particular context of this study (e.g., SSI problems in the context of the COVID-19 pandemic). As such, we suggest that researchers should take consideration of modifying our three-phase PL model based on their disciplinary context and particular background. Nonetheless, this study provides empirical evidence that the three-phase professional learning model can be a promising approach to support pre-service teachers' SSI-based instruction. Our three-phase model can be applicable and localized for pre-service teachers in other disciplines or in other countries.

**Author Contributions:** Conceptualization, P.H.; Methodology, P.H.; Formal analysis, M.H.; Investigation, M.H.; Data curation, M.H.; Writing—original draft, M.H.; Writing—review & editing, P.H. All authors have read and agreed to the published version of the manuscript.

**Funding:** This research was funded by the China Scholarship Council grant number 202008110028 And The APC was funded by the China Scholarship Council.

**Institutional Review Board Statement:** The authors declare that research with human subjects was conducted in accordance with the ethical principles under the Capital Normal University. We also declare that approval by local research ethics committee was not required since all participation of students and pre-service teachers was anonymous.

**Informed Consent Statement:** Informed consent was obtained from all subjects involved in the study.

**Data Availability Statement:** The data presented in this study are available on request from the corresponding author. The data are not publicly available due to the raw data are presented and analyzed in Chinese.

**Acknowledgments:** We would like to extend our heartfelt appreciation to the in-service teacher, Yi Gu, from the Affiliated High School of Beijing Jiaotong University, for her invaluable participation and assistance in completing the entire curriculum.

**Conflicts of Interest:** The authors declare that they have no competing interests.

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
