# Peer review of "Pre-Service Science Teachers’ Understanding of Socio-Scientific Issues Instruction through a Co-Design and Co-Teaching Approach Amidst the COVID-19 Pandemic"

_sustainability, doi:10.3390/su15108211_

Round 1

Reviewer 1 Report

I like idea of the authors, and I have only some small remarks for improvement of the paper. Generally it is acceptable, however I have some suggestions.

In the introduction it is sometimes difficult to tell whether a statement is based on the authors' opinion or whether it is something that comes from references or other sources of evidence. An example of such a text is in:

Lines 107 – 108: You wrote. “However, simultaneous curriculum design and teacher professional learning, 107 even for in-service teachers, is a demanding task.”

Rew: Make it clear whether it is your opinion or an assertion supported by references (evidence).

Methods:

I suggest that in some cases the authors take the position of the readers and help them to understand the procedure more easily. What may be well known to the authors may not necessarily be common knowledge. Example:

Lines 312 - 314. "In the first round, two authors used deductive codes based on "A Framework for Socio-scientific Issues Based Education" (Presley et al. 2013) to code the multiple data sources for each of the 12 features about SSI instruction."

As a minimum, include the 12 characteristics in parentheses or refer to the relevant section if they are mentioned elsewhere in a text (e.g., Figure 2).

Additionally, I believe that in-service teachers deserve to be mentioned by names in Acknowledgements.

Conclusions

Limitations of the study are not even mentioned. Indicate the future direction of follow-up studies.

Some details are missing, e.g. conflicts of interest, ethical concerns and the like.

References: Check the references carefully, because there are too many inconsistencies. Sometimes the name of the journal is missing, sometimes the volume or issue, etc. I recommend you check the reference style and ask an editor to clarify the problem (numbered style, use of abbreviations, cross-referencing via doi. etc.)

Reviewer 2 Report

The theme of the work is relevant. The abstract indicates the objective of the work

The method section is well structured.  Especially design, sampling and result parts are consistent with each other.

I think the manuscript meets expectations and is interesting (Understanding of

Socio-Scientific Issues Instruction through a Co-Design and

Co-Teaching Approach), well elaborated and structured. Some modifications are recommended to increase the final quality:

The discussion and conclusions and references section needs to be improved. In some part volume was written italic, some other issue was written italic.

Reviewer 3 Report

Thank you for the opportunity to review this interesting and practically useful manuscript.  I especially appreciated the incorporation of the co-design phase into the SSI instruction program.  This element is often overlooked when designing a program like this but can lead to major problem in terms of implementation if parties weren't collaborative during the planning. 

Here are suggested revisions: 

Readers would probably appreciate more information about parameters imposed on: (a) the selected topic, and (b) solutions to solve the socio-scientific "problem."  For instance, the topic appeared to be centered on halting transmission of COVID-19 through fomites rather than airborne mechanisms (an assumed boundary).  As another parameter, was emergency disinfection (do-it-yourself approaches) for use by households or small businesses (this assumption would likely shape designed solutions)?  In terms of problem-solving parameters, were certain disinfectants either discouraged or encouraged for students?  I'm not a chemist or virologist, but were methods such as ozone, UV light, alcohol, hydrogen peroxide, silver, and copper purposefully excluded?  Did the disinfectant need to work quickly or could it take hours to do its job?  It appeared that the methods ultimately used by students were similar but the "delivery system" varied (Figure 5); was this result by deliberate design or happenstance?          

Line 223: The authors should define what a "normal university" is.

Section 6 ("Discussion and Conclusion") is rather thin.  How can instructors in other disciplines outside of chemistry use this general approach to improve their own SSI instruction?  In other words, how generalizable do the authors believe their program is? (Lines 693-697 suggest little to no generalizability, but I believe that they could be bolder in claiming at least some degree of usefulness in other disciplines, even if the study at hand was based on only six pre-service teachers.)  The authors could expand on the final sentence (Lines 695-697) to further justify their earlier stated rationale for the importance of their study.  

As reported in this manuscript, the case study generated very "clean" findings, i.e., very few problems or complications were reported with respect to the 3-phase program, and student learning objectives were resoundingly met.  However, were there "outlying" findings or aspects that did not go as planned (or that were disappointing to the research team)?  Would other collaborative teams that included pre-service teachers benefit from learning about any difficulties that were encountered and how they were negotiated?  As one example from the manuscript, the authors wrote in Lines 616-621: "In the first lesson, students’ ideas on using disinfectants were scattered due to the fact that they had been exposed to too much extracurricular knowledge in a short period of time and had not paid close attention to their reading. PSTs noticed some students only focused on simple equations or principles they thought were easy to understand, but they were unable to apply them correctly. As a result, when asked why they chose a particular principle, most students were unable to provide a clear answer and were confused."  To better exemplify the problem noted within this direct quote, could the authors also provide a specific example or two from their data?

Generally very effective.

Reviewer 4 Report

Dear Authors,

Congratulations, you have been able to submit articles to this journal. Overall your article is quite good, but there are some parts that need to be explained more so that it is perfect. All of these notes are in the articles we reviewed. Thank you very much, I hope your article will be published successfully through this article.

Best regards

Dear Editors,

Thank you for trusting me to review this article. Overall the English in your article is quite good, but there are some parts that need to be explained more so that it is more perfect. All of these notes are in the articles we reviewed. Thank you very much, for the trust.

Best regards
